# Effects of Different Molecular Weight Oxidized Dextran as Crosslinkers on Stability and Antioxidant Capacity of Curcumin-Loaded Nanoparticles

**DOI:** 10.3390/foods12132533

**Published:** 2023-06-29

**Authors:** Dongyan Shen, Hongzhou Chen, Mingwei Li, Ling Yu, Xiangfei Li, Huawei Liu, Qiaobin Hu, Yingjian Lu

**Affiliations:** 1College of Food Science and Engineering, Nanjing University of Finance and Economics/Collaborative Innovation Center for Modern Grain Circulation and Safety, 3 Wenyuan Road, Nanjing 210023, China; dongyanshen1996@163.com (D.S.); liquidmirtop1@163.com (M.L.); lingyu20180626@126.com (L.Y.); xiangfeili@nufe.edu.cn (X.L.); liuhuawei@njau.edu.cn (H.L.); 2Anhui Guotaizhongxin Testing Technology Co., Ltd., 22nd Floor, Huishang Square, Hefei 230041, China; 3College of Health Solutions, Arizona State University, 850 N 5th Street, Phoenix, AZ 85004, USA

**Keywords:** curcumin, modified chitosan, oxidized dextran, nanoparticles, stability, antioxidant capacity

## Abstract

Curcumin is a polyphenolic compound that has been widely investigated for its health benefits. However, the clinical relevance of curcumin is limited due to its low water solubility and inefficient absorption. Therefore, curcumin is often encapsulated in nanocarriers to improve its delivery and function. In this study, composite nanoparticles composed of stearic acid-modified chitosan (SA-CS) and sodium caseinate (NaCas) were formed using sodium periodate-oxidized dextran with different molecular weights as a crosslinking agent. The effects of oxidized dextran (Odex) with different molecular weights on the composite nanoparticles were compared. The optimal SA-CS/NaCas/Odex composite nanoparticle (NPO) was obtained using an Odex (150 kDa)-to-SA-CS mass ratio of 2:1. Its size, polydispersity index (PDI), and zeta potential (ZP) were 130.2 nm, 0.149, and 25.4 mV, respectively. The particles were highly stable in simulated gastric fluid (SGF) in vitro, and their size and PDI were 172.3 nm and 0.263, respectively. The encapsulation rate of NPO loaded with curcumin (Cur-NPO) was 93% under optimal ultrasonic conditions. Compared with free curcumin, the sustained release of Cur-NPO significantly reduced to 17.9%, and free-radical-scavenging ability improved to 78.7%. In general, the optimal prepared NPO exhibited good GI stability and has potential applications in the formulation of orally bioactive hydrophobic drugs.

## 1. Introduction

Curcumin is a polyphenolic plant component that has been widely studied for its antioxidation and health benefits [1]. However, due to its low solubility, curcumin has low oral bioavailability and is poorly absorbed by the intestinal tract. Hence, curcumin is often encapsulated in nanocarriers to improve its delivery to tissues [2].

Chitosan (CS) has good biocompatibility and antibacterial, anticancer, and adhesion properties [3], and sodium caseinate (NaCas) has a strong affinity for both hydrophilic and hydrophobic functional groups and has been widely used in nanomaterials [4]. Zhang et al. (2023) reported that naringenin-zein-sodium caseinate-galactosylated chitosan nanoparticles could exhibit a better lipid-lowering effect [5]. CS composite nanoparticles prepared via ionic gel synthesis are also promising encapsulation and delivery systems for bioactive compounds [6,7]. However, composite nanoparticles are readily degraded by enzymes and rapidly released into the gastrointestinal (GI) tract. Moreover, although chemical crosslinking agents are commonly used to harden protein composite nanoparticles, they are non-food-grade, limiting their use in foods [8].

Previous studies have shown that dextrin has good potential as a nano-delivery material [9]. Odex is prepared by oxidating dextrin and has been extensively studied as a nontoxic crosslinker for nanoscale systems [10]. Odex can also form Schiff base crosslinks by reacting with the amino groups of proteins and thereby stabilize nanostructures [10,11,12]. Odex has been used as a crosslinking agent in gels and has shown the best behavior for potential applications in wound healing or soft tissue regeneration [12]. The polymer composition and surface characteristics of composite nanoparticles determine their internal stability and intestinal absorption, which means that using the correct crosslinking agent is vital. Yan et al. (2023) studied that the highly cross-linked hydrogel could release curcumin in a more controlled manner compared to the case using low-molecular-weight Odex [13]. Previous studies have discussed the antioxidant activity and immunomodulatory activity of dextran with different molecular weights, and it is expected that dextran with a 40 kDa molecular weight may have greater potential in these two aspects [14]. However, there have been few studies on characteristics of Odex crosslinked with varying molecular weights.

Therefore, we used various molecular weights of Odex to SA-CS and NaCas composite nanoparticles. The effects of molecular weight and concentration on the resulting composite nanoparticles were systematically evaluated by in vitro GI stability testing. In addition, the ultrasonic parameters for curcumin encapsulation were optimized to study the loading capacity and controlled release profile of composite nanoparticles as a potential oral delivery system.

## 2. Materials and Methods

### 2.1. Materials

CS (50–190 kDa with a deacetylation degree of 79.0%), NaCas, 2,2′-azino-bis(3-ethylbenzothiazoline-6)-sulfonic acid (ABTS), curcumin, pepsin, and pancreatin were purchased from Sigma-Aldrich (St. Louis, MO, USA). SA, ascorbic acid, and acetic acid were obtained from Aladdin Biochemical Technology Co. (Shanghai, China). Dextran samples with various molecular weights (10, 40, 70, 100, and 150 kDa) were purchased from Macklin (Shanghai, China). All other chemicals (including anhydrous ethanol) were of analytical grade and were obtained from Sinopharm Group Co., Ltd. (Shanghai, China).

### 2.2. Preparation of SA-CS

SA-CS was synthesized via an ascorbic acid/hydrogen peroxide (H_2_O_2_) redox reaction under a nitrogen atmosphere, following a previously reported method with some modifications [15,16]. First, a 10 mg/mL solution of CS was prepared by dissolving CS in an aqueous solution of acetic acid (1% *v*/*v*). Fifty mL of this CS solution was treated with 1 mL of aqueous H_2_O_2_ (1 M) containing 54 mg of ascorbic acid, and the resulting mixture was magnetically stirred for 30 min at room temperature. SA was dissolved in ethanol to form a 0.1 mg/mL working fluid, 50 mL of which was added to the CS prepared above. The resulting mixture was allowed to react in the dark for 24 h under a continuous flow of nitrogen. After this, the mixture was heated at 70 °C in a water bath for 6 h to remove ethanol. Finally, the reaction solution was lyophilized. The resulting dry SA-CS powder was characterized via FTIR spectroscopy (Thermo Scientific IS5, Waltham, MA, USA).

### 2.3. Preparation of Odex

Odex samples with different molecular weights were prepared via a previously described method [17] with slight modifications. First, 2 g of dextran with various molecular weights (10, 40, 70, 100, and 150 kDa) was dissolved in 40 mL ultrapure water. The resulting solution was treated with 3.52 g of sodium periodate, and then the mixture was stirred in the dark for 6 h. The reaction mixture was then placed into an 8-14 kDa molecular weight cutoff dialysis tube and dialyzed in ultrapure water for 72 h, with ultrapure water being replaced every 8 h. Finally, the reaction solution was lyophilized. The resulting dry Odex products were characterized via FTIR spectroscopy.

### 2.4. Preparation of NPs

#### 2.4.1. Optimization of the SA-CS/NaCas Mass Ratio

SA-CS/NaCas composite nanoparticles (NPs) were prepared according to the ionic gelation of CS. First, SA-CS and NaCas were separately dissolved to form SA-CS stock solutions with a concentration of 2 mg/mL. Then, four 5 mL aliquots of NaCas solutions (of various concentrations) were added to five 5 mL aliquots of SA-CS solution to give solutions with SA-CS-to-NaCas mass ratios of 1:1, 2:1, 3:1, and 4:1. These mixtures were stirred for 30 min, and then heated in a water bath at 80 °C for 30 min [11]. The particle size, PDI, and ZP were determined via dynamic light scattering (DLS) on a Malvern Zetasizer Nano ZS (Malvern Instruments, Ltd., Hitachi, Totyo, Japan).

#### 2.4.2. Optimization of the Crosslinking Process

The composite nanoparticles with the optimal SA-CS/NaCas mass ratios were selected for further optimization of the cross-linking process. Odex samples with various molecular weights were dissolved in ultrapure water to form 20 mg/mL solutions. SA-CS and NaCas solutions were mixed in optimal mass ratios and then stirred for 30 min, then 1 mL aliquots of Odex solution to give solutions with Odex-to-SA-CS mass ratios of 0.5:1, 1:1, 1.5:1, and 2:1. Each molecular weight is added according to the above mass ratio. The resulting mixtures were heated at 80 °C on a water bath for 30 min to form SA-CS/NaCas/Odex composite nanoparticle (NPO) [11]. Odex was replaced with an equal volume of ultrapure water as a control. The particle size, ZP, and PDI of NPOs were determined by DLS to reveal the influence of different crosslinkers.

The stability of these NPOs in a simulated GI tract was investigated using simulated gastric fluid (SGF) and simulated intestinal fluid (SIF) [10]. First, 100 µL of freshly prepared NPO was combined with 900 µL of SGF (pH = 2, containing 1 mg/mL pepsin), and the resulting mixture was incubated at 37 °C for 2 h. Similarly, 100 µL of freshly prepared NPO was combined with 900 µL of SIF (pH = 7, containing 10 mg/mL pancreatin), and the resulting mixture was incubated at 37 °C for 4 h. At the end of the reaction, the samples were taken out. Particle size and PDI were regarded as indicators of the stability of crosslinking agents in NPOs in GI fluids in vitro.

### 2.5. Curcumin Encapsulation

#### 2.5.1. Optimization of Loading Rate

A literature method based on pre-dissolved NaCas was modified and supplemented with ultrasonic treatment [18]. An appropriate volume of a pre-dissolved solution of curcumin was directly added to four solutions of NaCas containing 2.5%, 5%, 7.5%, and 10% loadings (by weight) of nanoparticles, respectively, and the resulting mixtures were stirred at room temperature for 10 min. Then, the reaction mixture was sonicated at 350 W for 90 s and then added slowly to 5 mL SA-CS. The resulting mixtures were stirred for 30 min and then treated with an appropriate amount of the optimized crosslinker. The resulting mixtures containing crosslinker were stirred and then heated in a water bath at 80 °C for 30 min to form curcumin-loaded nanoparticles, which were subsequently characterized by the physicochemical properties (particle size, PDI, and ZP) were determined by DLS, as above.

The curcumin encapsulation ability of NPO was determined as follows [17]. Samples were centrifuged at a low speed (5000 rpm) for 10 min. The resulting precipitates were dissolved in anhydrous ethanol and diluted to an appropriate concentration, and the curcumin concentration of these dilutions was determined at 430 nm. The loading capacity (*LC*) and encapsulation efficiency (*EE*) were calculated as follows:(1)LC (%) = (Total Cur−Free Cur)/(Total mass of NPs)×100%
(2)EE (%)=(Total Cur−Free Cur)/(Total Cur)×100%
where “Cur” denotes curcumin, and “Total mass of NPs” is the total mass of polysaccharide (SA-CS), protein (NaCas), and curcumin.

#### 2.5.2. Optimization of Ultrasonic Power

The effects of ultrasonic power on the properties and embedding efficiency of NPO were investigated. First, the ultrasonication power parameters were optimized at a fixed optimal loading rate (see Section 2.5.1). Then, to further improve the embedding efficiency, the optimal ultrasonic power (250, 350, 450, or 550 W) was determined. The remaining steps were identical to those in Section 2.5.1.

#### 2.5.3. Optimization of Ultrasonication Duration

The effects of ultrasonication duration on the properties and embedding efficiency of NPO were also investigated. The ultrasonication duration was varied (45, 90, 135, 180 s) with the ultrasonic power fixed at the optimal condition (see Section 2.5.2). The remaining steps were identical to those in Section 2.5.1.

### 2.6. FTIR Analysis

All samples were analyzed via FTIR spectroscopy. The speed and scan ranges were 4 cm^−1^ and 500-4000 cm^−1^, respectively, and the data were processed in OriginPro 8.0.

### 2.7. Transmission Electron Microscopy Analysis

The morphologies of the sample under various conditions were observed via transmission electron microscopy (TEM; HT7800, Hitachi, Japan). Each sample was diluted to afford a 1 mg/mL test sample. Three microliters of each test sample were carefully dropped onto a carbon-coated TEM grid and allowed to sit for 2 min. After this time, the grid was washed with 100 µL of 0.5% phosphotungstic acid. The morphology of the stained sample was then captured by TEM.

### 2.8. Fluorescence Analysis

The fluorescence intensity of curcumin-loaded nanoparticles was measured using a fluorescence spectrophotometer (F-7000; Hitachi, Tokyo, Japan). The emission spectra in a quartz cell with a 10 mm path length were recorded at an excitation wavelength of 425 nm from 450 to 700 nm. Curcumin solution was pre-dissolved in anhydrous ethanol to afford a 4 mg/mL solution. A control solution containing unencapsulated curcumin was diluted with ultrapure water and anhydrous ethanol at the same concentration as Cur-NPO. In addition, the same concentration of curcumin-casein was prepared.

### 2.9. Kinetic Release Profile of Curcumin

The kinetics of curcumin release in the nanoparticles was determined via dialysis [19]. To create precipitation conditions for curcumin, the release medium was prepared by premixing equal volumes of ethanol, SGF, or SIF [20]. First, 3 mL of curcumin-loaded nanoparticles or free curcumin was placed in an 8-14 kDa molecular weight cut-off dialysis tube. The loaded tube was then immersed in 60 mL of SGF, and this setup was gently shaken for 2 h at room temperature. Afterward, the loaded tube was transferred to 60 mL of SIF and shaken for 4 h at room temperature. At predetermined time points, 1 mL of the dialysis medium was withdrawn, and an equivalent volume of fresh medium was replenished. The concentration of curcumin in the dialysate was determined at 430 nm using a microplate reader (BioTek Instruments, Inc.; Winooski, VT, USA).

### 2.10. ABTS Free-Radical-Scavenging Assay

The ABTS free-radical-scavenging assay was used to determine the antioxidant activity of curcumin-loaded nanoparticles, according to the method of Xie et al. (2014) [21]. Briefly, the curcumin-loaded nanoparticles were dissolved and used to generate a concentration series (0.005, 0.01, 0.02, 0.03, 0.04, and 0.05 mg/mL). Then equal volumes of 7 mM ABTS and 4.95 mM potassium persulfate were dissolved in ultrapure water, and the mixture was stored in the dark for 12 h. The solution of ABTS free radicals was diluted with sufficient PBS (0.2 M, pH 7.4) to afford a working solution with an absorbance of 0.7 ± 0.02 at 734 nm. Then, 200 µL of this working solution and 20 µL of each sample were combined in a 96-well microplate, and the absorbance of the samples was read at 734 nm on a microplate reader. A solution containing the same concentration of free curcumin was formed with anhydrous ethanol (control 1) and water (control 2) as the two control samples of the nanoparticles [21].
(3)$$\mathrm{ABTS}\;\mathrm{free}\;\mathrm{radical}\;\mathrm{scavenging}\;\mathrm{activity}\;(\%)=[1-(A_{s}-A_{b})/A_{c}]\times 100\%$$
where *A*_s_, *A*_b_, and *A*_c_ are the absorbance of the sample, background, and control, respectively.

### 2.11. Statistical Analysis

All data analyses were performed in triplicate, and the data are represented as means ± standard deviations (S.D). A one-way analysis of variance was performed using Duncan’s multiple comparison tests to compare the differences between the results by SPSS 16.0. The significance level *p* was set to 0.05.

## 3. Results and Discussion

### 3.1. NPO Preparation

#### 3.1.1. Optimization of the SA-CS/NaCas Mass Ratio

The electrostatic attraction between the positively charged SA-CS and the negatively charged NaCas, and the hydrophobic interactions between the SA segment of the species on the CS and the hydrophobic residues on NaCas are the main driving forces for the formation of polymeric composite nanoparticles [10]. Previous studies have shown that the mass ratio of polysaccharides to protein plays an important role in the formation of the composite nanoparticle. Therefore, the effects of the SA-CS/NaCas mass ratio on the nanoparticle physicochemical properties were studied. The particle size, PDI, and ZP of a series of SA-CS/NaCas composite nanoparticles with different mass ratios of SA-CS to NaCas (i.e., 1:1, 2:1, 3:1, and 4:1) were analyzed. The particle size increased with the mass ratio (Figure 1) because after saturation of the SA-CS binding sites, more SA-CS became involved in the crosslinking process owing to the excess NaCas. In contrast, the PDI and ZP decreased with an increasing mass ratio. The decrease in the PDI was possibly due to the cross-linking of SA-CS with NaCas, increasing the density and homogeneity structure. Regarding the ZP, the initial solutions were positively charged owing to the presence of SA-CS, but as the concentration of NaCas (which is negatively charged) in the solution increased, the net positive charge on the surface of solutions decreased [22]. The changing trend of particle size, potential, and PDI of composite nanoparticles is consistent with the study of gallic acid-modified chitosan-sodium caseinate nanoparticles [11].

Yu et al. (2016) showed that protein could provide a relatively stable environment for curcumin and improve its dispersal in solution. Thus, we expected that the nanoparticle structure should contain as much NaCas as possible [23]. To test the improvement of the hydrophobic active material curcumin embedded later. Composite nanoparticles with an excellent particle size (<150 nm) and surface potential (>30 mV) were formed at SA-CS/NaCas mass ratios of 3:1 and 4:1, but as the low NaCas content in the two ratios disfavored curcumin embedding. Moreover, the smaller the composite nanoparticles, the higher their ZP and the greater their loading capacity and curcumin encapsulation efficiency. In particular, composite nanoparticles with an SA-CS/NaCas mass ratio of 2:1 was smaller and had a higher ZP than those with an SA-CS/NaCas mass ratio of 1:1. Therefore, the mass ratio of 2:1 was used for further study, and the content of protein in the system is improved, which is superior to the composite nanoparticles prepared by the previous study [11].

#### 3.1.2. Optimization of the Crosslinking Process

Crosslinkers such as Odex ensure the stability of colloids, and the particle size and surface properties of nanoparticles determine their stability in vivo. The size, PDI, and ZP varied with Odex’s molecular weight and content. With increasing Odex content, the particle size of all samples first decreased and then increased (Figure 2A,B). In the presence of certain concentrations of Odex, the nanoparticles’ structure was more compact and had a smaller particle size, owing to ionic interactions between Odex and the SA-CS amino group. Similarly, Previous research reported that the particle size of their nanoparticle first increased and then decreased with increasing crosslinker content [24]. However, further increases in the Odex content generated thick surface coatings on nanoparticles that increased their particle size, which is attributable to polymeric flexibility and high hydrophilicity [25]. Therefore, there is a correlation between nanoparticle diameter and Odex content. The nanoparticles containing 150 kDa Odex had the lowest variation in size.

Odex can influence ZP by covering their surface, and the ZP of NPO (Odex-treated) was less than the surface potential of the control (Odex-free) (+32 mV; Figure 2C), indicating that Odex was adsorbed by the nanoparticle. Furthermore, the ZP decreased with increasing Odex content, which indicates that the ZP was negatively correlated with their Odex content. All molecular weight crosslinking agents showed the same potential change trend. This is consistent with the results of previous studies [26]. The PDI values ranged from 0.1 to 0.2 (<0.3), indicating that the nanoformulations were uniformly dispersed (i.e., not aggregated) in an aqueous solution.

The stability, particle size, and PDI of each sample incubated in SGF and SIF were measured. The control groups degraded in SGF (Table 1). At pH 2, the particle size and PDI increased from 153.1 to 318.5 nm and 0.148 to 0.424, respectively. SA-CS did not protect NaCas from low pH- or pepsin-mediated degradation, resulting in the breakdown of the nanostructures. At pH 7, the particle size and PDI increased more significantly, from 153.1 to 840.9 nm and 0.148 to 0.550, respectively, because the p*K*_a_ values of amino groups on SA-CS are close to pH 7 and underwent very little protonation, which triggered severe aggregation [10]. Therefore, improved crosslinkers must be developed to prevent their enzymatic degradation under GI conditions.

Odex can stabilize them as a crosslinker for proteins or polysaccharides [27,28]. However, its effects on the stability of composite nanoparticles in simulated GI fluids varied considerably with its molecular weights. At pH 2, nanoparticles crosslinked with Odex of different molecular weights were significantly more stable than the control. In particular, composite nanoparticles crosslinked with 150 kDa Odex had particle sizes change in the range of 42.1 to 49.3 nm at all concentrations. The sizes change the range of NPOs containing 10, 40, 70, and 100 kDa Odex were 40.6–61.3 nm, 40.6–53.4 nm, 37.6–70.9 nm, and 44.5–68.3 nm, respectively. The smaller the change in size, the more stable the NPOs structure and the more tolerant the NPOs tolerance of an acidic environment [29]. In addition, the 40, 70, or 150 kDa Odex content of NPO was positively correlated with their nanostructural stability in SGF. However, this was not so for the 10 and 100 kDa Odex, which may be related to the complex SGF environment. Moreover, the PDIs of NPOs were less than that of the control (0.424), demonstrating that the former was not prone to aggregation or settlement and, thus, more stable than the latter. The PDI was not correlated with their Odex concentration or the molecular weight of Odex. At pH 7, NPO with an Odex (150 kDa)-to-SA-CS mass ratio of 2:1 showed excellent stability, and their size and PDI (259.6 nm and 0.311, respectively) were significantly better than those of the control (840 nm and 0.550, respectively). The sample prepared under other conditions and incubated in SIF for four hours exhibited precipitation (the corresponding phase data are not shown in Table 1). This behavior is attributable to the crosslinks with low molecular weights and, in low concentrations, cannot effectively protect nanoparticles from aggregation and precipitation [10]. The optimal NPO was obtained using an Odex (150 kDa)-to-SA-CS mass ratio of 2:1.

### 3.2. Encapsulation of Curcumin

#### 3.2.1. Optimization of Drug Loading

Curcumin has numerous useful biological activities but has low aqueous solubility and, thus, low bioavailability [30,31,32,33]. Recent studies have confirmed proteoglycan composite nanoparticles to be excellent carriers for hydrophobic substances. In this study, we combined simple, fast, and low-cost ultrasonic techniques to synthesize nanoparticles with high curcumin-encapsulation efficiencies. The ultrasonic preparation of Cur-NPO is illustrated in Figure 3. The optimal process was determined in terms of curcumin encapsulation efficiency and size, PDI, and ZP.

The curcumin-encapsulation efficiencies were determined under curcumin loadings of 2.5%, 5.0%, 7.5%, and 10.0% (Figure 4A). The initial concentration of curcumin was inversely proportional to the curcumin-encapsulation efficiency of NPO: the encapsulation efficiency at a loading of 10% was the lowest (61.1%), owing to oversaturation causing a constant decrease in the NaCas concentration [33]. Figure 4A,B shows the effects of curcumin loading on the properties of Cur-NPO. The particle size increased with curcumin loading, consistent with the results of previous studies on the encapsulation of curcumin [34,35]. The particle sizes and PDIs of Cur-NPO treated with 2.5%, 5.0%, or 7.5% of curcumin were slightly greater than those of the blank nanoparticles (curcumin-free), demonstrating the Cur-NPO effectively encapsulated, bound, and dispersed curcumin binding capacity. In particular, Cur-NPO treated with 7.5% curcumin exhibited a slightly higher but acceptable PDI (0.248 < 0.3). However, Cur-NPO treated with 10% curcumin exhibited a considerably higher particle size (311.5 nm) and PDI (0.635) than Cur-NPO treated with lower proportions of curcumin, indicating that curcumin that was not protein-bound was poorly dispersed in the aqueous phase, which was echoed by these low curcumin-encapsulation efficiency.

Although a high curcumin-encapsulation efficiency was the main goal, the stability of the Cur-NPO construct was also important. Considering the results, a 7.5% loading of curcumin was selected for the next optimization step.

#### 3.2.2. Optimization of Ultrasonication Power

The curcumin-encapsulation efficiencies of Cur-NPO treated at ultrasound frequencies of 250, 350, 450, and 550 W were determined (Figure 4C). With increasing ultrasound power, the curcumin-encapsulation efficiency first increased and then decreased. The highest curcumin-encapsulation efficiency (93%) was obtained at 450 W. However, 550 W ultrasound treatment negatively affected the properties of NaCas, probably because of the higher energy caused overfolding of casein, leading to the exposure of a large proportion of hydrophobic residues to the solution. This would have increased the formation of insoluble nanoparticle aggregates, thereby decreasing the nanoparticles’ curcumin-encapsulation efficiency [36]. In addition, ultrasound power has a small but measurable effect on nanoparticle properties: with increasing ultrasound power, the particle size and PDI slightly decreased, while their ZP fluctuated (Figure 4C,D). However, all the samples exhibited good stability. Considering these results, we selected an ultrasound power of 450 W for the next experiment.

#### 3.2.3. Optimization of Ultrasonication Duration

The curcumin-encapsulation efficiencies of Cur-NPO were determined under ultrasonication durations (Figure 4E). With increasing ultrasonication duration, the curcumin-encapsulation efficiency first increased and then decreased. The highest curcumin-encapsulation efficiency (93%) was obtained at 90 s. This suggests that this ultrasonication duration best facilitated the unfolding of NaCas molecules, which enhanced their hydrophobic interactions with and the encapsulation of curcumin. However, at longer ultrasonication durations, excessive energy was applied-analogous to high-energy ultrasonication—and hydrophobic areas would have been excessively exposed to the aqueous solution, resulting in the formation of insoluble aggregates, thereby decreasing the efficiency of encapsulation of curcumin, a hydrophobic substance [37]. Furthermore, with increasing ultrasonication duration, the particle size and PDI first decreased and then increased while ZP fluctuated (Figure 4E,F). This suggests that the optimal ultrasonication duration favored the formation of small, homogeneous Cur-NPO.

Floris et al. (2013) found that an increased duration of ultrasonication led to a decrease in the number of nanoparticles formed. In the current study, prolonged ultrasonication negatively affected nanoparticle properties [38]. Chen et al. (2022) also reported that the encapsulation efficiency of the nanoparticle dispersion systems after ultrasonic treatment was higher than without ultrasonic treatment [39]. In particular, Cur-NPO ultrasonicated for 180 s exhibited a particle size and PDI of 250.7 nm and 0.573, respectively. This is consistent with the concept of “overprocessing” proposed by Desrumaux (2002), who reported that particle size increased at high ultrasound power. In the current study, Cur-NPO ultrasonicated for 90 s exhibited good stability and a curcumin-encapsulation efficiency of up to 93%, which was significantly higher than those ultrasonicated for other durations [40]. Thus, we used an ultrasound power of 450 W and an ultrasonication duration of 90 s for the preparation of Cur-NPO.

### 3.3. FTIR Analysis

FTIR revealed the interaction between composite nanoparticles and modified materials or curcumin and the microstructural differences between the components (Figure 5A). The spectra of native CS featured characteristic peaks at 1648, 1589, and 1312 cm^−1^, which correspond to the C=O stretching vibration of the amide I bands, the N−H bending vibration of amide II bands, and the C−N stretching vibration of amide III bands [41,42]. The spectra of the synthesized SA-CS exhibited C=O stretching peaks and N-H bending peaks at 1628 and 1543 cm^−1^, respectively. The peaks were shifted compared with those of the native sample and exhibited a different intensity; this suggests that the SA was conjugated to the amino and hydroxyl groups of the CS [43]. Furthermore, the SA-CS spectra exhibited new peaks at 1698 cm^−1^, corresponding to C=C stretching, indicating the formation of ester bonds [44]. The characteristic peaks of NaCas were observed at 1640, 1535, and 3284 cm^−1^, corresponding to amide I and amide II and O-H stretching, respectively.

The formation of Odex (150 kDa) is illustrated in Figure 3A. The new peak at 1728 cm^−1^ in the Odex (150 kDa) spectrum corresponds to the C=O stretching of the aldehyde group, indicating successful Odex (150 kDa) preparation. Moreover, the spectra of the optimal prepared NPO featured characteristic peaks at 1647, 1543, and 1728 cm^−1^, corresponding to amide I, amide II, and ester bonds, respectively. These peaks were highly similar to those of SA-CS, NaCas, and Odex, and the changes in the wavenumber in the amide bond region indicate the occurrence of electrostatic interactions between biopolymers [45]. All the composite nanoparticles prepared with Odex exhibited peaks at 1037-1643 cm^−1^, corresponding to the stretching of the Schiff base formed during crosslinking [46]. Furthermore, NPO exhibited only one band at 1728 cm^−1^, possibly due to the overlapping of the ester bond-extended group in SA-CS with the unreacted aldehyde group in Odex [44]. Composite nanoparticles containing Odex with different molecular weights exhibited similar spectra (Appendix A).

As shown in Figure 5A, curcumin features several characteristic FTIR peaks at 3511, 1627, 1509, and 1427 cm^−1^, corresponding to the O-H telescopic vibration of phenol, a ketone group, and an olefinic C-H bending vibration, respectively [47]. The characteristic −C=C− peaks of curcumin at 963 and 812 cm^−1^ were not present in the spectrum of Cur-NPO, indicating that curcumin was encapsulated, consistent with the conclusion by Liu et al. (2017). Furthermore, the characteristic peak of curcumin at 1280 cm^−1^, attributable to the C-O tensile vibration of the benzene ring, weakened after curcumin was encapsulated by the NPO. This indicates that intermolecular hydrophobic and hydrogen-bond interactions possibly altered the structures of some of curcumin’s functional groups [47].

### 3.4. Morphology of Prepared Nanoparticles

The morphology of composite nanoparticles prepared by Odex crosslinking with the same concentration but with different molecular weights and Cur-NPO were obtained through TEM analyses. All the composite nanoparticles exhibited similar spherical morphologies (Figure 6A–H). The addition of Odex and ultrasonication treatment had little effect on these morphologies. The TEM-determined particle size of all samples was less than the DLS-measured size; this was probably due to dehydration and shrinkage during the preparation. These findings are similar to those of Zhang et al. (Y.L. Zhang, Dou, and Jin 2010) [48].

### 3.5. Fluorescence Analysis

The molecular interactions between curcumin and NPO were further analyzed via fluorescence spectroscopy. Curcumin dissolved in anhydrous ethanol showed a strong fluorescence peak at 542.2 nm (Figure 5B). Compared with the fluorescence intensity of free curcumin in water, the fluorescence intensity of curcumin encapsulated in NPO was enhanced, and blue shifted to 518 nm. This indicates that the curcumin structure or conformation was altered upon encapsulation, resulting in complex formation [49]. The same phenomenon has been observed [50]. A curcumin-casein mixture also exhibited a blue shift and enhanced fluorescence intensity, indicating that curcumin bonded with NaCas via hydrophobic interactions, consistent with the findings of others [11,51].

### 3.6. Kinetic Release Profile of Curcumin

Free curcumin and Cur-NPO were controllably released in SGF and SIF. Free curcumin was rapidly released in SGF and SIF (Figure 7), with a final release rate of 63.3%. A previous study reported continuous release of free curcumin, with a final release rate of 68.4% [11]. Yang et al. studied that the total amount of Cur released after six hours was 70.04 %, and the amount of Cur released under SIF (43.99 %) was significantly greater than that under SGF (26.05 %) [52]. However, the final release rate of Cur-NPO was 17.9%, much less than that of free curcumin (63.3%). The results showed that Cur-NPO had a better retention effect. This was likely because curcumin was encapsulated within NPO nuclei by hydrophobic interactions, and its release was greatly limited by the dense structure of composite nanoparticles. Rodriguez et al. (2019) also reported a slow release for maize protein/NaCas complex nanoparticles [29].

### 3.7. In Vitro Antioxidant Analysis (ABTS)

The antioxidant activities of Cur-NPO, NPO (without curcumin), and controls (curcumin in water and aqueous ethanol) were determined via ABTS free-radical clearance analysis (Figure 8). With increasing curcumin concentration, from 0.005 to 0.050 mg/mL, the clearance rate of free radicals from Cur-NPO (compared to curcumin in water) increased. The low solubility of curcumin possibly limited the number of dissolved molecules present to react with free radicals [49], so the antioxidant activity was only 22.3% at the highest curcumin concentration. However, the antioxidant activity of Cur-NPO in water was up to 78.7% greater than controls of curcumin in water and aqueous ethanol. This is attributable to NPOs’ improved dispersion of curcumin in water, resulting in greater radical scavenging [11,42,44,49].

## 4. Conclusions

In this study, Odex with various molecular weights had a remarkable effect on Cur-NPO. Odex with 150 kDa added to Cur-NPO and combined with the application of ultrasound made Cur-NPO achieve smaller particle size, PDI. Furthermore, Odex with 150 kDa could effectively improve encapsulation to 93%. Cur-NPO also exhibited higher in vitro antioxidant activity than free curcumin and could encapsulate and release hydrophobic molecules in a sustained manner. Our findings suggest that Odex with various molecular weights had a positive effect on the stability and cumulative release ability of nanoparticles in SGF and SIF, which could have potential applications in encapsulation systems. However, Cur-NPO needed further studies on cytotoxicity and uptake direction in order to provide the basis for possible cell experimental research.

## Figures and Tables

**Figure 1 foods-12-02533-f001:**
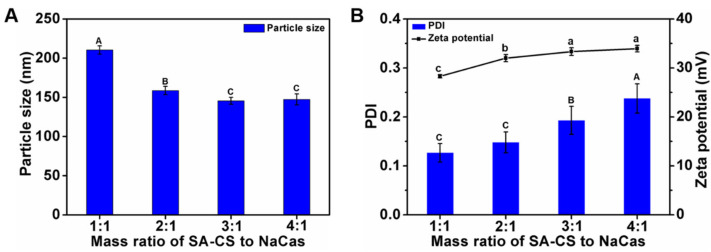
Effects of mass ratio of SA-CS to NaCas on particle size (**A**); PDI and zeta potential (**B**) of composite nanoparticles. The different letter represents a significant difference of different concentrations under the same molecular weight compared with the control group (*p* < 0.05).

**Figure 2 foods-12-02533-f002:**
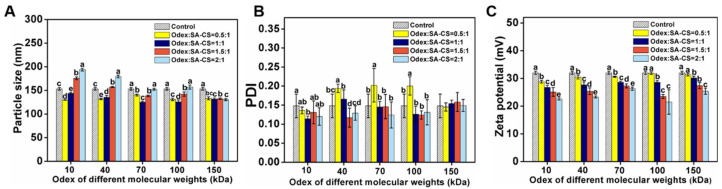
Effect of the proportion of Odex addition prepared from each molecular weight on particle size (**A**); PDI (**B**); and zeta potential (**C**) of NPOs. The different letter represents a significant difference of different concentrations under the same molecular weight compared with the control group (*p* < 0.05).

**Figure 3 foods-12-02533-f003:**
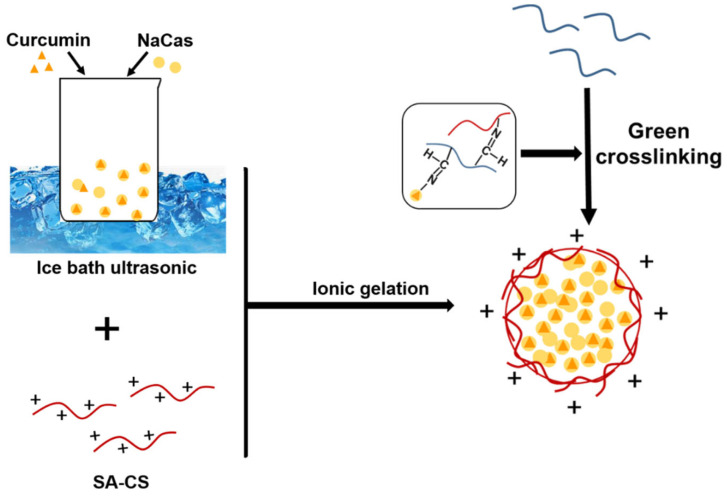
Proposed mechanism of preparation of curcumin-loaded SA-CS/NaCas/Odex nanoparticles (Cur-NPO).

**Figure 4 foods-12-02533-f004:**
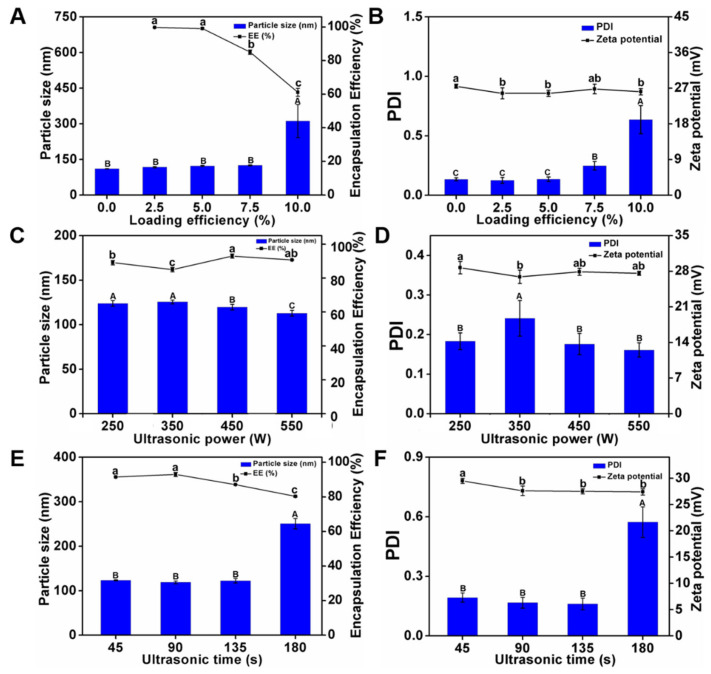
Characteristics of Cur-NPO with different curcumin loading efficiency: (**A**) particle size and encapsulation efficiency; (**B**) PDI and zeta potential; Characteristics of Cur-NPO with different ultrasonic power: (**C**) particle size and encapsulation efficiency; (**D**) PDI and zeta potential; Characteristics of Cur-NPO with different ultrasonic time: (**E**) particle size and encapsulation efficiency; (**F**) PDI and zeta potential. The molecular weight of Odex is 150 kDa. The different letter represents a significant difference of different concentrations under the same molecular weight compared with the control group (*p* < 0.05).

**Figure 5 foods-12-02533-f005:**
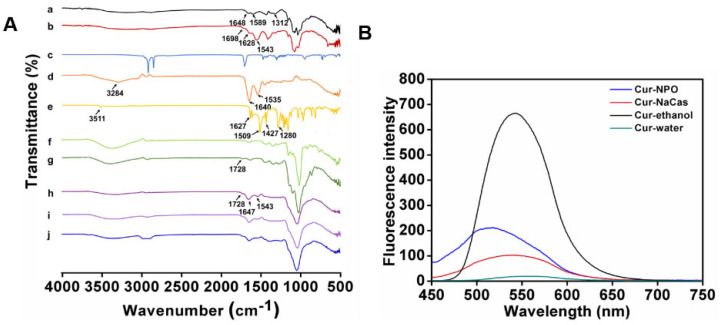
Characterization of nanoparticles: (**A**) FT-IR spectrum of individual biopolymer and various nanoparticles prepared under different conditions. a. CS; b. SA-CS; c. SA; d. NaCas; e. Curcumin; f. dextran (150 kDa); g. Odex (150 kDa); h. NPO (150 kDa) without ultrasound; i. NPO (150 kDa); j. Cur-NPO (150 kDa). (**B**) The intrinsic fluorescence property of free curcumin and Cur-NPO (150 kDa).

**Figure 6 foods-12-02533-f006:**
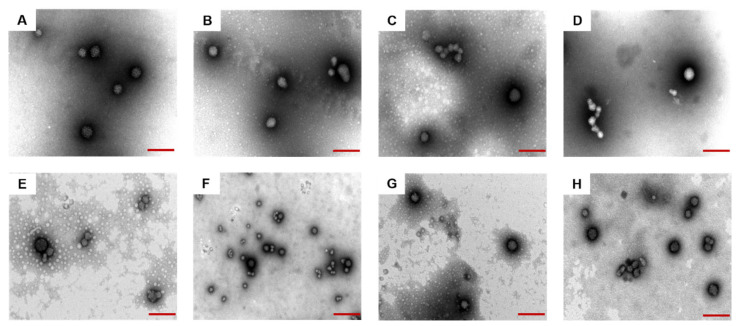
TEM images of prepared nanoparticles under different conditions (**A**) NP (Odex-free); (**B**) NPO (10 kDa); (**C**) NPO (40 kDa); (**D**) NPO (70 kDa); (**E**) NPO (100 kDa); (**F**) NPO (150 kDa) without ultrasound; (**G**) NPO (150 kDa); (**H**) Cur-NPO (150 kDa). Scale bars are 200 nm.

**Figure 7 foods-12-02533-f007:**
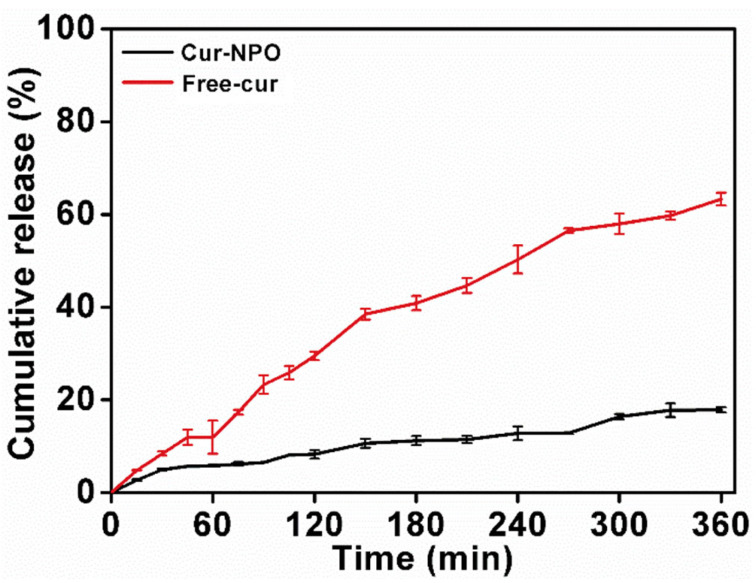
The kinetic release profile of free curcumin and Cur-NPO under different simulated gastric and intestinal conditions.

**Figure 8 foods-12-02533-f008:**
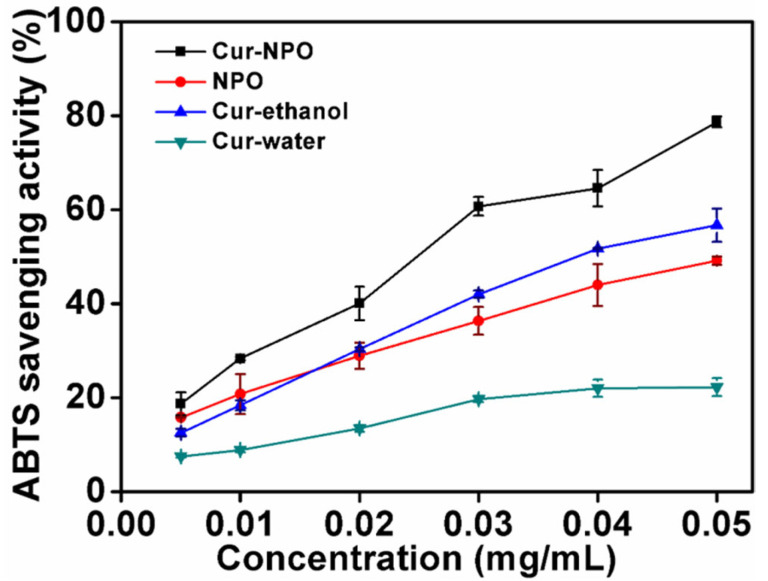
Scavenging activities against ABTS radicals of NPO without loading curcumin and Cur-NPO. Two controls were prepared by diluting a pre-dissolved curcumin stock solution with aqueous ethanol and water.

**Table 1 foods-12-02533-t001:** Stability of nanoparticles with different proportions of Odex crosslinking procedures in SGF.

Samples	Odex/SA-CS (*w*/*w*)	pH = 2.0	
		Size (nm)	PDI
Control		318.5 ± 24.4 ^a^	0.424 ± 0.059 ^a^
Odex (10 kDa)	0.5:1	176.3 ± 3.3 ^d^	0.247 ± 0.005 ^b^
	1:1	184.3 ± 9.6 ^d^	0.221 ± 0.026 ^c^
	1.5:1	236.9 ± 12.2 ^c^	0.246 ± 0.023 ^c^
	2:1	239.4 ± 12.1 ^b^	0.225 ± 0.043 ^c^
Odex (40 kDa)	0.5:1	185.2 ± 7.8 ^d^	0.340 ± 0.032 ^b^
	1:1	175.9 ± 4.2 ^d^	0.264 ± 0.011 ^c^
	1.5:1	204.4 ± 6.7 ^c^	0.236 ± 0.011 ^c^
	2:1	221.2 ± 7.3 ^b^	0.271 ± 0.034 ^c^
Odex (70 kDa)	0.5:1	211.0 ± 9.0 ^b^	0.369 ± 0.011 ^b^
	1:1	184.9 ± 1.8 ^c^	0.338 ± 0.028 ^bc^
	1.5:1	187.5 ± 4.1 ^c^	0.304 ± 0.030 ^c^
	2:1	190.0 ± 6.9 ^c^	0.254 ± 0.015 ^d^
Odex (100 kDa)	0.5:1	199.1 ± 7.6 ^c^	0.346 ± 0.015 ^b^
	1:1	178.8 ± 3.3 ^d^	0.350 ± 0.015 ^b^
	1.5:1	186.6 ± 4.0 ^cd^	0.277 ± 0.012 ^c^
	2:1	213.5 ± 2.6 ^b^	0.307 ± 0.036 ^c^
Odex (150 kDa)	0.5:1	182.5 ± 1.0 ^b^	0.287 ± 0.023 ^b^
	1:1	175.4 ± 2.9 ^b^	0.286 ± 0.033 ^b^
	1.5:1	178.8 ± 2.4 ^b^	0.251 ± 0.018 ^b^
	2:1	172.3 ± 2.0 ^b^	0.263 ± 0.022 ^b^

Values represent mean ± standard deviation, and the different letter represents a significant difference of different concentrations under the same molecular weight compared with the control group (*p* < 0.05).

## Data Availability

Data is contained within the article or Appendix A.

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
