# Peer review of "Effects of Different Molecular Weight Oxidized Dextran as Crosslinkers on Stability and Antioxidant Capacity of Curcumin-Loaded Nanoparticles"

_foods, 2023, doi:10.3390/foods12132533_

Round 1
Reviewer 1 Report
I have evaluated the manuscript (Foods-2401001) titled “Effects of Different Molecular Weight Oxidized Dextran as Crosslinkers on Stability and Antioxidant Capacity of Curcumin-Loaded Nanoparticles” by Lu and coworkers. I found this article interesting for the readers and followed the journal Foods’ scope. All standard methods were used for the study with proper discussion of background of the topic of the research.
I would recommend this article be published in the journal Foods after minor corrections.
The author needs to address the following comments/corrections.
1. The author needs to rewrite the equations 1-3 with proper font size and footnotes.
2. Is there is any effect of pH on controlled released of free curcumin and Cur–NPO in SGF and SIF?
3. Any stability data is available for Cur–NPO in GI in different pH?
4. What is the control used for antioxidant analysis/or scavenging activities of Cur–NPO?
5. The author could move some parts of results and discussion to the supplementary information to make this manuscript more interesting.
6. The author could include the following relevant references.
(a) Wasiak I, Kulikowska A, Janczewska M, Michalak M, Cymerman IA, Nagalski A, Kallinger P, Szymanski WW, Ciach T. Dextran Nanoparticle Synthesis and Properties. PLoS One. 2016 Jan 11;11(1):e0146237. doi: 10.1371/journal.pone.0146237. PMID: 26752182; PMCID: PMC4713431.
(b) Laura G. Gómez-Mascaraque, José Alberto Méndez, Mar Fernández-Gutiérrez, Blanca Vázquez, Julio San Román,. Oxidized dextrins as alternative crosslinking agents for polysaccharides: Application to hydrogels of agarose–chitosan, Acta Biomaterialia,Volume 10, Issue 2, 2014,Pages 798-811.
(c) Soeiro VC, Melo KR, Alves MG, Medeiros MJ, Grilo ML, Almeida-Lima J, Pontes DL, Costa LS, Rocha HA. Dextran: Influence of Molecular Weight in Antioxidant Properties and Immunomodulatory Potential. Int J Mol Sci. 2016 Aug 19;17(8):1340. doi: 10.3390/ijms17081340. PMID: 27548151; PMCID: PMC5000737.
Author Response
Thank you for your valuable comments. We respond to the above questions one by one.
- The author needs to rewrite the equations 1-3 with proper font size and footnotes.
Answer: Thank you for your kind advise. We have modified the format of the formula.
- Is there is any effect of pH on controlled released of free curcumin and Cur–NPO in SGF and SIF?
Answer:Thank you for your kind advise. pH has very little effect on Cur-NPO nanoparticles both in SGF and SIF. While pH has significant effect on the release of free curcumin.
- Any stability data is available for Cur–NPO in GI in different pH?
Answer: Thank you for your kind advise. Normally, the pH of SGF fluctuates between 1 and 3, and there are periods when the pH is 2. So we chose to simulate gastric juices at a pH of 2.
- What is the control used for antioxidant analysis/or scavenging activities of Cur–NPO?
Answer: Thank you for your kind advise. The controls included curcumin in water and aqueous ethanol, NPO was used to describe the ability of components other than curcumin to clear ABTS.
- The author could move some parts of results and discussion to the supplementary information to make this manuscript more interesting.
Answer: Thank you for your kind advise. We summarized and condensed the conclusions and discussions of the research.
- The author could include the following relevant references.
Answer: Thank you for your kind advise. We have inserted these references in introduction.

Reviewer 2 Report
The authors studied the effects of different molecular weight oxidized dextran as crosslinkers on stability and antioxidant capacity of curcumin-loaded nanoparticles. Prepared nanoparticles of curcumin were characterized physicochemically and evaluated for the stability and antioxidant effects. Overall, the work is interesting and will be beneficial for the fellow researchers. I will recommend it after major revisions:
Abstract: The results about drug release kinetics and antioxidant activity are completely missing in the abstract. The authors are advised to include some quantitative results of drug release studies and antioxidant activity in the abstract.
Symbols, units, subscripts and superscripts: The authors are advised to present all units in SI system and there should be a space between the physical quantity and unit. All the symbols should be italics and their subscripts or superscripts should be non-italics throughout the manuscript.
Introduction: The introduction is obsolete. The information about the target drug, polymer, chitosan, and polymer/chitosan-based nanoparticles needs significant improvement. In addition, the authors have not discussed the existing drug delivery systems of curcumin. Therefore, discuss the existing drug delivery systems of curcumin and write the advantages of present drug delivery system over its existing drug delivery systems. Authors are also suggested to add recent literature. You can consult the following articles to make this manuscript more useful to the readers:
Methods: Authors are advised to include literature for each experimental procedure.
Figure 5: The letters in FTIR spectra are not readable. The wave numbers are presented on x-axis so you can remove these values from the figure.
Results and discussion: The authors are advised to discuss the results in the light of previous studies. Please compare your results with previous studies and mention clearly how your work is important in comparison to already been reported.
Authors are advised to include the main limitation of work at the end of results and discussion section and just before the conclusion.
Avoid abbreviations before giving their explanation.
Conclusion: The conclusion should be concise and to the point indicating the application of the work.
Minor editing of English language required.
Author Response
Thank you for your valuable comments. We respond to the above questions one by one.
- Abstract: The results about drug release kinetics and antioxidant activity are completely missing in the abstract. The authors are advised to include some quantitative results of drug release studies and antioxidant activity in the abstract.
Answer: Thank you for your kind advise. We have adjusted the description in the abstract section, adding the data in vitro simulation (SGF) and ABTS scavenging capacity.
- The authors are advised to present all units in SI system and there should be a space between the physical quantity and unit. All the symbols should be italics and their subscripts or superscripts should be non-italics throughout the manuscript.
Answer: Thank you for your kind advise. We examined the corresponding formulas and made improvements to them.
- The introduction is obsolete. The information about the target drug, polymer, chitosan, and polymer/chitosan-based nanoparticles needs significant improvement. In addition, the authors have not discussed the existing drug delivery systems of curcumin. Therefore, discuss the existing drug delivery systems of curcumin and write the advantages of present drug delivery system over its existing drug delivery systems. Authors are also suggested to add recent literature. You can consult the following articles to make this manuscript more useful to the readers:
Answer: Thank you for your kind advise. We made some changes in the abstract, introduction, and conclusion to make this manuscript more useful to readers.
Methods: Authors are advised to include literature for each experimental procedure.
Answer: Thank you for your kind advise. We changed it in the right places to make it easier for readers to read.
Figure 5: The letters in FTIR spectra are not readable. The wave numbers are presented on x-axis so you can remove these values from the figure.
Answer: Thank you for your kind advise. We have changed picture 5 to make it more clear and concise.
Results and discussion: The authors are advised to discuss the results in the light of previous studies. Please compare your results with previous studies and mention clearly how your work is important in comparison to already been reported.
Answer: Thank you for your kind advise. We added some comparisons with recent studies to the discussion to reflect the characteristics of our research.
Authors are advised to include the main limitation of work at the end of results and discussion section and just before the conclusion.
Answer: Thank you for your kind advise. We have made modifications in the corresponding section.
Avoid abbreviations before giving their explanation.
Answer: Thank you for your kind advise. We've adjusted the placement of abbreviations so that the ones that don't make sense don't appear in the right place.
Conclusion: The conclusion should be concise and to the point indicating the application of the work.
Answer: Thank you for your kind advise. We condensed our conclusions to better illustrate the characteristics of our study. And some discussion of potential problems.

Reviewer 3 Report
This work provides a non-toxic and food-grade method for improving the oral bioavailability of curcumin, which has potential applications in the formulation of hydrophobic drugs. however curcumin has been classified as both a PAINS (pan-assay interference compounds) and an IMPS (invalid metabolic panaceas) candidate. This classification raises concerns about the reliability of curcumin's activity in vitro and in vivo. Please add more disscusion about the issues of curcumin such as why curcumin not a drug like compound and how to overcome poor bioavailability problem.
Author Response
Thank you for your valuable comments. We respond to the above questions one by one.
Comments and Suggestions for Authors
This work provides a non-toxic and food-grade method for improving the oral bioavailability of curcumin, which has potential applications in the formulation of hydrophobic drugs. however curcumin has been classified as both a PAINS (pan-assay interference compounds) and an IMPS (invalid metabolic panaceas) candidate. This classification raises concerns about the reliability of curcumin's activity in vitro and in vivo. Please add more disscusion about the issues of curcumin such as why curcumin not a drug like compound and how to overcome poor bioavailability problem.
Answer: Thank you for your kind advise. We add some discussion about curcumin in some parts of the paper, compare it with some recent studies, and finally discuss some potential problems that may exist in the study in the conclusion.

Round 2
Reviewer 2 Report
The authors have addressed the previous concerns. The revised manuscript is suitable for publication in its present form.